# Association Between Family Caregiver Burden and Affiliate Stigma in the Families of People with Dementia

**DOI:** 10.3390/ijerph17082772

**Published:** 2020-04-17

**Authors:** Jian-An Su, Chih-Cheng Chang

**Affiliations:** 1Department of Psychiatry, Chang Gung Medical Foundation, Chiayi Chang Gung Memorial Hospital, Chiayi 613044, Taiwan; jian.7715@gmail.com; 2School of Medicine, Chang Gung University, Taoyuan 33302, Taiwan; 3Department of Nursing, Chang Gung Institute of Technology, Taoyuan 33303, Taiwan; 4Department of Psychiatry, Chi Mei Medical Center, Tainan 70246, Taiwan; 5Department of Health Psychology, Chang Jung Christian University, Tainan 71101, Taiwan

**Keywords:** dementia, caregiver, caregiver burden, affiliate stigma

## Abstract

Family caregivers of people with dementia (PWD) have a heavy care burden. Affiliate stigma is the stigma internalized by individuals associated with PWD. Limited research has addressed the affiliate stigma among caregivers of PWD and its influence on caregiver burden. Thus, our study investigated the burden of caregivers of PWD and its relationship with affiliate stigma. In addition, we examined the factors related to affiliate stigma. This cross-sectional study was conducted in a general hospital in Taiwan. We recruited 270 PWD and their family caregivers from the outpatient department. Relevant demographic and clinical assessment data of the patients and caregivers were evaluated. Regression analysis was performed to examine the factors associated with affiliate stigma. In total, 23.7% of the family caregivers had depression and 37.4% had anxiety. Male caregivers had higher levels of anxiety and heavier care burdens related to affiliate stigma compared with female caregivers. Moreover, characteristics such as younger age and low levels of dependence in daily activities among PWD were associated with increased affiliate stigma. A higher family caregiver burden was related to more severe affiliate stigma. Interventions for decreasing the family caregiver burden might reduce the effect of affiliate stigma.

## 1. Introduction

According to the report of Alzheimer’s Disease International (ADI) in 2019, the incidence of dementia is increasing rapidly. Currently, 50 million people globally are estimated to have dementia. This number is expected to triple by 2050 [1]. Alzheimer’s disease affects not only individuals with the disease but also their caregivers. According to the annual report of ADI, approximately half of the family caregivers of people with Alzheimer’s stated that their health, work, and social life had suffered. Many factors have been reported to affect the perceived burden in caring for family members with dementia. The factors influencing the aforementioned burden include those related to people with dementia (PWD), such as disease severity, behavior problems, and disease duration [2,3,4], as well as factors related to caregivers, such as kinship, gender, coping strategies, and supporting resources [2,3]. Studies have also indicated that caregivers of PWD are more stressed than the family members of PWD. These caregivers are vulnerable to chronic disease and mental illness, such as depression and anxiety [5,6].

In 1963, Erving Goffman defined stigma as “an attribute, behavior, or reputation which is socially discrediting in a particular way: it causes an individual to be mentally classified by others in an undesirable, rejected stereotype rather than in an accepted normal one [7].” Stigma involves the processes of labeling, negative stereotyping, separation, discrimination, emotional reaction, and status loss [8]. Researchers have considered the importance of stigma awareness in the field of dementia. In the online survey–based report of ADI, 75% of PWD responded that they experienced negative association and 40% of them reported being treated negatively, including losing friends and being isolated [9]. Approximately one in four PWD attempted to conceal their diagnosis from others due to the stigma associated with the disease.

The influence of stigma is observed not only in PWD but also in their caregivers, relatives, and associated health professionals. The stigma among caregivers, relatives, and health professionals is referred to as “courtesy stigma” or “stigma by association,” which indicates that negative behaviors of the public are directed toward those closely associated with PWD [7,10,11]. People closely affiliated with a stigmatized individual may be affected by the public stigma that prevails in society. Courtesy stigma is not the only stigma related to the associates of PWD. The associates of PWD may also develop affiliate stigma, which refers to the internationalization of negative public views toward oneself [12]. Affiliate stigma associated with Alzheimer’s disease has three key components: cognitive attributions (neglectful caregiver and punishment from God), emotional reactions (anger, fear and shame) and behavioral reactions (concealment, isolation, and not seeking help) [13]. This internalization may lead to serious consequences, such as lower self-esteem and self-efficacy, feelings of hopelessness, and poor quality of life according to the stigma studies with family caregivers of people with mental illness [14,15,16,17]. Studies have indicated that family caregivers who perceive stigmatization experience high stress, negative emotions, a high care burden, and social isolation; they engage in less help-seeking behaviors and may be unable to provide high-quality care [18,19,20,21,22,23].

The aforementioned studies addressing the stigma in family caregivers of PWD have considered family stigma as stigma by association. However, few studies have been conducted on affiliate stigma in family caregivers of PWD. Moreover, the influence of caregiver burden on affiliate stigma is not completely understood. Thus, this study investigated the relationship between the caregiver burden in family caregivers of PWD and affiliate stigma as well as the demographic and clinical factors contributing to this stigma type.

## 2. Materials and Methods

### 2.1. Study Design

A cross-sectional design was used in this study, and all the study participants were recruited from a general hospital in Taiwan. All the caregiver participants had one family member aged older than 65 years with any type of dementia and were the caregivers of the affected family members. The diagnosis of dementia was based on the diagnostic criteria of the Diagnostic and Statistical Manual of Mental Disorders, Fourth Edition (DSM-IV) [24]. The inclusion criteria for caregiver participants were as follows: at least 20-years-old; able to understand, speak, and can read Mandarin Chinese or Taiwanese; and willing to sign informed consent after being provided with a complete explanation. Those who did not meet the inclusion criteria were excluded. In addition, we also collected the information of PWD from their medical record and their caregiver including their demographic data and the report of psychological assessment after PWD and their family caregiver signed the informed consent. We would get the consent simultaneously from their legally authorized representatives if the patient had impaired capacity to give consent after assessment by their treating psychiatrists or the principal investigator (also a psychiatrist).

### 2.2. Assessment

The Caregiver Burden Inventory (CBI) was developed by Novak and Guest [25] and translated into Chinese by Chou et al. [26]. It comprises four dimensions for measuring subjective burden: emotional burden, social burden, time-dependent burden, and physical burden. The aforementioned scale comprises 24 items, which are rated on a five-point Likert scale. The higher the CBI score is, the higher is the level of caregiver burden.

The affiliate stigma scale, which was developed by Mak and Cheung [12] and validated by Chang [27], was used in this study to assess the internalized stigma of caregivers of family members with a mental illness in Taiwan [28]. This instrument comprises three domains: cognitive, affect, and behavioral domains. It consists of 22 items, which are rated on a four-point Likert scale. The higher the mean score on the Affiliate Stigma Scale is, the higher is the level of affiliate stigma.

The Taiwanese Depressive Questionnaire (TDQ) comprises 18 items for assessing depressive symptoms in the previous 1 week. These items are also rated on a 4-point Likert scale. A TDQ score higher than 19 indicates severe depressive symptoms. The TDQ is culturally relevant and has satisfactory reliability and validity in Taiwanese people [29].

The Beck Anxiety Inventory (BAI) [30] was used to assess the severity of anxiety. The BAI comprises 21-items, which are scored on a scale of 0 (not at all) to 3 (severely). A BAI score of more than eight indicates the presence of anxiety. The BAI has been translated to Taiwanese Mandarin with excellent internal consistency and validity [31].

The information collected from participants included their demographic data, relationship with family members having dementia, living condition, years of education, and employment status. Information was also collected on whether the participants shared their care burden with others. The patient data included demographic information, illness duration, first visit to a psychiatrist or neurologist due to dementia symptoms, and employment status.

Other assessments included Neuropsychiatric Inventory (NPI) score [32,33], Barthel Index (BI) [34], Clinical Dementia Rating (CDR) [35], and the Mini-Mental Status Examination (MMSE) score [36].

The study protocol was approved by the Institutional Review Board of Chang Memorial Hospital(102-3378B), and informed consent was obtained from all the participants.

### 2.3. Statistical Analysis

A hierarchical regression model was used to analyze the factors contributing to affiliate stigma. First, the caregivers’ demographic data were included in Model 1. Clinical data regarding the PWD, such as NPI score, BI, CDR, and MMSE score, were added in Model 2. Finally, the caregiver burden, anxiety, and depression scores of the caregivers were added in Model 3.

## 3. Results

### 3.1. Descriptive Statistics of the Main Variables

A total of 270 participants and their family members with dementia were recruited in this study. The sociodemographic data of the caregivers are presented in Table 1. The mean age of the caregivers was 52.3 ± 12.2, and the gender distribution was approximately equal. Most of the caregivers lived with their family members with dementia, and half of the caregivers had full- or part-time jobs. The daily caring time was 10.1 ± 9.3 h, and the mean duration of caregiving was 29.7 ± 29.4 months. Most of the participants (83.7%) were primary caregivers, and 76.3% shared the caregiving burden with others. The most common kinship for caregiving was child (61.9%), followed by spouse (13.3%).

The mean caregiver burden score was 40.5 ± 19.1 in the CBI. A total of 23.7% of the participants had depressive symptoms and 37.4% had at least mild anxiety, as assessed using the TDQ and BAI, respectively.

For the patients with dementia recruited in our study (Table 2), the mean age was 79.0 ± 6.3 years old. Most of the patients with dementia were female (64.4%), and 96.7% of the patients were not employed or had retired. Dementia severity, BI scores, and neuropsychiatric symptoms are presented in Table 3. Most of the patients had mild dementia according to the CDR, and the mean MMSE score was 13.2 ± 5.6.

### 3.2. Hierarchical Regression Analyses

Factors related to affiliate stigma were identified using hierarchical regression analyses with caregivers’ socio-demographics, patients’ socio-demographics and clinical data, and caregivers’ psychological and stigma-related factors controlled (Table 4). None of the caregivers’ demographics except male gender were significantly associated with affiliate stigma. In regard to patients’ socio-demographics and clinical variables, age and Barthel Index score of patients with dementia were significantly related to affiliate stigma score. After caregivers’ psychological and stigma-related factors were taken into account, anxiety and caregiver burden were significant related to affiliate stigma. These indicate that greater affiliate stigma was associated with male gender, high levels of anxiety, high caregiver burden, and being a caregiver who had the ill relatives with younger age and low levels of dependence in daily activities. In the sample, the overall model explained 53% of the variance in the score of affiliate stigma. 

## 4. Discussion

This study examined the relationship between caregiver burden and affiliate stigma as well as the factors related to PWD and their caregivers that affected affiliate stigma. The results indicated that male sex, high anxiety levels, and high caregiver burden were the factors related to caregivers that affected affiliate stigma. Moreover, younger age and low functional dependence in daily activities were the factors related to PWD that predicted affiliate stigma.

Previous studies have focused on the affiliate stigma associated with severe mental illnesses such as schizophrenia, bipolar disorder, and major depressive disorder. The results of these studies indicate that affiliate stigma may lead to unhappiness, helplessness, and negative emotions [12,37,38]. Limited research has focused on affiliate stigma in caregivers of PWD. Studies on stigma in dementia caregivers have mostly addressed family stigma by assessing it with the Family Stigma in Alzheimer’s Disease Scale (FS-ADS). This scale assesses stigma perception, stigma experience, and structural stigma, which are not specific to affiliate stigma [39]. Further studies should be conducted to address the influence of affiliate stigma on caregivers of PWD.

One study conducted in the United States presented a positive correlation between caregiver burden and stigma [40]. Another study conducted in Israel reported that caregiver stigma, as assessed using the FS-ADS, is a predictor of caregiver burden [41]. This study indicated that caregiver burden is a significant predictor of affiliate stigma. Considering the aforementioned results, the relationship between caregiver burden and stigma might be reciprocal, and these factors might have a bidirectional influence. Greater affiliate stigma may make caregivers perceive a greater sense of subjective burden. In addition, heavy caregiver burden may result in caregivers’ guilty feeling because they think that they do not provide good enough care to the patients. This process including cognitive, emotional and behavior reactions could give rise to higher levels of affiliate stigma.

A stigma-related study conducted by Kahn among patients with memory disorder revealed that female caregivers experienced higher levels of stigma and care burden than male caregivers did [40]. However, in this study, male caregivers had higher levels of affiliate stigma than female caregivers did. This disagreement may have been due to several reasons. First, the stigma measures considered in this study and Kahn’s study were different. We only focused on affiliate stigma, whereas Kahn used the FS-ADS, which encompasses broader stigma concepts, including stigma perception and structural stigma. Second, differences in kinship between the two studies may have caused different results. The most common kinship in this study was adult child, whereas the main kinship in Kahn’s study was spouse. Third, cultural differences may have contributed to the discrepancy between the results in the aforementioned studies. In Taiwan, women are seen as the main caregivers, especially for older adults [42,43]. Male caregivers in Taiwanese society may accept the caring role to a lesser extent because of social expectations.

Younger age and low functional dependence in daily activities were the factors related to the PWD that predicted high levels of affiliate stigma. The results reflected a real-world phenomenon that although PWD face the problem of poor recent memory, they can still appropriately perform most daily activities, such as self-care, moving within the community, and shopping for necessities, at a younger age or in the early stages of dementia. However, the quality of self-care and social interaction may deteriorate for PWD, and they may easily have disagreements with others or get lost. People unfamiliar with the behavior of PWD might view these patients as “psychotic.” People may blame family caregivers for providing a poor home environment or mismanaging ill relatives [11]. In the long term, caregivers may internalize negative stereotypes from the general public, which may result in affiliate stigma. In addition, PWD who are relatively independent in their daily activities or are of younger age are less likely to be considered as having dementia by the general public. Therefore, the families of these individuals might be unable to accept the dementia diagnosis, which might give rise to affiliate stigma.

Some studies have indicated that dementia severity as well as the behavioral and psychological symptoms of dementia (BPSD) are highly related to caregiver burden and might influence affiliate stigma. However, in this study, the aforementioned factors did not exhibit a significant association with affiliate stigma in the final model. This result can be explained by the fact that most of the PWD recruited in this study were in the early stage of dementia and thus, the influence of severity of dementia was small. BPSD was a significant predictor for affiliate stigma in Model 2; however, this effect was counteracted by adding the variable of caregiver burden. This result imply that an association exists between affiliate stigma and caregiver burden but not between affiliate stigma and BPSD. Further larger-scale studies should be conducted for patients having different dementia severity levels to verify the association between dementia severity, BPSD, and affiliate stigma.

Caregivers of PWD have been reported to be vulnerable to depression and anxiety, which are highly related to caregiver burden [2,44]. In this study, caregivers’ anxiety, rather than depression, was significantly associated with affiliate stigma. Affiliate stigma may exert negative impacts on the caregivers, which lead to greater anxiety. In another way, caregivers with significant anxious symptoms may be more sensitive to public views and tend to internalize the negative perceptions against themselves. The effect of depression may be masked by caregiver burden because of the relationship between depression and caregiver burden [45]. Limited research has examined the effects of both depression and caregiver burden on affiliate stigma. Further large-scale studies are required to examine this issue.

This study has certain limitations. First, all the PWD were recruited from one general hospital in southern Taiwan; therefore, the results cannot be generalized to the entire population of Taiwan. Second, the assessment of anxiety and depression was based on self-report questionnaires, which might be unable to reflect the real severity of symptoms. Third, in Chinese culture, caregivers of PWD may be concerned about “losing face” [46,47], which may be an important factor related to affiliate stigma. Furthermore, perceived family stigma and positive aspects of caregiving may play a part in affiliate stigma among family caregivers [48]. However, we did not assess these variables in our study. Future studies are warranted to examine the relationship among affiliate stigma and those factors. Fourth, our cross-sectional research design limited the inference on causal effects between affiliate stigma and relevant factors such as caregiver burden and anxiety. The advantage of this study is that the quantitative method, rather than the qualitative method used by most similar studies, was used to investigate affiliate stigma. In addition, we collected demographic and clinical information on both caregivers and patients. To the best of our knowledge, no previous study has considered the aforementioned information for both caregivers and patients.

## 5. Conclusions

This study indicates that caregiver burden and other predictors are significantly associated with affiliate stigma. The study results suggest that strategies designed for reducing caregiver burden may also reduce affiliate stigma and its significant impact. In particular, such strategies can be first targeted toward male caregivers with high levels of anxiety and their relatives with dementia with a younger age at onset and low levels of functional dependence in their daily living. Future studies may focus on the development of effective intervention strategies for reducing affiliate stigma among family caregivers of PWD.

## Figures and Tables

**Table 1 ijerph-17-02772-t001:** Socio-demographic characteristics of family caregivers (*n* = 270).

	*n* (%)	Mean ± SD
Age (years)		52.3 ± 12.2
Years of education		11.3 ± 4.2
Daily contact time with patients (hours)		10.1 ± 9.3
Duration of caregiving (months)		29.7 ± 29.4
Gender		
Male	128 (47.4)	
Female	142 (52.6)	
Marital status (married)	206 (76.3)	
Employment status		
Full-time/Part-time employment	144 (53.3)	
Housekeeper/Retired/No employment	126 (46.7)	
Religion(yes)	235(87.0)	
Relationship with patient		
Parent	1 (0.4)	
Spouse	36 (13.3)	
Child	167 (61.9)	
Other	66 (24.4)	
Living with patient (yes)	193 (71.5)	
Primary caregiver (yes)	226 (83.7)	

**Table 2 ijerph-17-02772-t002:** Socio-demographic characteristics of people with dementia (*n* = 270).

	*n* (%)	Mean ± SD
Age (years)		79.0 ± 6.3
Years of education		3.4 ± 4.4
Dementia onset (age)		75.6 ± 12.2
First time psychiatric treatment (age)		75.7 ± 7.3
Gender		
Male	96 (35.6)	
Female	174 (64.4)	
Marital status		
Married	147 (54.4)	
Separated/Divorced/Widowed/Single	123 (45.6)	
Employment status		
Full-time/Part-time employment	9 (3.3)	
Retired/No employment	261 (96.7)	
Religion (yes)	260 (96.3)	
Ever psychiatric hospitalization (yes)	14 (5.2)	

**Table 3 ijerph-17-02772-t003:** Clinical profiles of people with dementia.

	*n* (%)	Mean ± SD
Neuropsychiatry Inventory		23.2 ± 23.2
Barthel Index		70.1 ± 33.3
Mini-Mental State Examination		13.2 ± 5.6
Clinical Dementia Rating		
0.5	89 (33.0)	
1	117 (43.3)	
2	55 (20.4)	
3	8 (3.0)	
4	1 (0.4)	

**Table 4 ijerph-17-02772-t004:** Factors associated affiliate stigma analyzed by hierarchical regression model.

	Model 1			Model 2			Model 3		
	B	SE B	ẞ	B	SE B	ẞ	B	SE B	ẞ
**Caregivers’ socio-demographics**									
Age	0.00	0.00	−0.06	0.00	0.00	0.05	0.00	0.00	0.03
Gender (Ref: Male)	0.04	0.07	0.04	0.01	0.07	0.01	−0.15	0.05	−0.15 **
Marital status (Ref: Married)	−0.12	0.08	−0.11	−0.08	0.08	−0.07	−0.09	0.06	−0.08
Years of education	0.00	0.01	0.03	0.01	0.01	0.12	0.00	0.01	−0.03
Employment status (Ref: Full-time/Part-time employment)	−0.02	0.07	−0.02	−0.02	0.07	−0.02	−0.02	0.05	−0.02
Share the care (Ref: No)	−0.15	0.08	−0.13	−0.09	0.08	−0.08	0.06	0.06	0.05
**Patients’ socio-demographics and clinical data**									
Age				−0.02	0.01	−0.21 **	−0.01	0.00	−0.14 **
Gender (Ref: Male)				0.05	0.07	0.04	0.03	0.05	0.03
Marital status (Ref: Married)				0.04	0.07	0.04	0.07	0.05	0.07
Neuropsychiatry Inventory				0.01	0.00	0.33 ***	0.00	0.00	0.07
Barthel Index				0.00	0.00	0.05	0.00	0.00	0.17 **
Clinical Dementia Rating				0.00	0.07	0.00	0.03	0.06	0.04
Mini-Mental State Examination				−0.01	0.01	−0.05	0.00	0.01	0.04
**Caregivers’ Psychological and stigma-related factors**									
Anxiety							0.01	0.00	0.17 *
Depression							0.00	0.00	0.07
Caregiver burden							0.02	0.00	0.57 ***
R^2^ (Adjusted R^2^)	0.02 (−0.01)			0.17 (0.12)			0.53 (0.50)		
F-value (*p*)	0.77 (0.60)			3.72 (<0.001)			16.97 (<0.001)		

* *p* < 0.05; ** *p* < 0.01; *** *p* < 0.001.

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
