# Peer review of "Association Between Family Caregiver Burden and Affiliate Stigma in the Families of People with Dementia"

_ijerph, 2020, doi:10.3390/ijerph17082772_

Round 1

Reviewer 1 Report

Abstract

Would be useful to define the term caregiver given authors also mentioned other caring roles such as health professionals - either in the abstract or introduction

Introduction

Clear and well written definition of courtesy and affiliate stigma

p2 line 46, stereotypes can also be positive, important to mention the process of stigmatization involves "negative" stereotypes

p2 line 60, references 13-16 are mental health references, although consequences may resonate for carers of PWD, perhaps acknowledge that this body of work is predominantly mental health based rather than studies in carers of pwd

Qualitative work by Werner and colleagues on affiliate stigma can be cited her to provide a picture of what affiliate stigma represents for carers

Methods

Participants were caregivers however later in the results (p4 line 126), it is stated that PWD were also recruited to the study but the procedure does not mentioned how they were consented (e.g. capacity criteria) or whether this information was obtained through the caregiver/hospital.

Description of actual data collection procedure is omitted in the methods

Results

Section 3.2 is hard to read, it may benefit from shorter and more precise sentences where the factors relating to the caregiver and PWD respectively are more obvious to the reader.

Discussion

It may be plausible to suggest that perceived family stigma and positive aspects of caregiving may play a part in the overall affiliate stigma experienced by carers although this was not measured in the current study worth considering the effect - see Family Stigma Instrument (Mitter et al., 2018 in BJPsych Open) adaptation of the affiliate stigma scale

Reviewer 2 Report

Important research to support this population of patients and caregivers. Well written and easy to read.

My main concern is with the findings where you indicate that caregiver burden "predicts" affiliate stigma. Does it? or does a person feel affiliate stigma which then creates a greater sense of caregiver burden? Statements such as "This study indicates that caregiver burden and other predictors contribute to affiliate stigma." and this "Anxiety might increase feelings of stigma."  - does it? or does the stigma increase anxiety? Should the researcher be on the lookout for caregivers who are more likely to have affiliate stigma to address the likelihood that those caregivers will have greater anxiety and caregiver burden? Instead of looking for those who have high caregiver burden and addressing the affiliate stigma with interventions? 

I do not believe the direction of the associations has been well supported by previous theory or theoretical models. It would be helpful if you expand on how the regression model was based on a theory for adding the concepts in a certain order.

Clarification of those statements or increasing the support would help. 
